# Managing Hospitalized Patients Taking SGLT2 Inhibitors: Reducing the Risk of Euglycemic Diabetic Ketoacidosis

**Julia Selwyn** [1,*] and **Ariana R. Pichardo-Lowden** [2]

1 Pennsylvania State Hershey Medical Center Department of Medicine and Pediatrics, Hershey, PA 17033, USA
2 Pennsylvania State Hershey Medical Center Department of Medicine, Division of Endocrinology, Diabetes, and Metabolism, Hershey, PA 17033, USA
* Correspondence: jselwyn@pennstatehealth.psu.edu or jselwyn@liberty.edu

**Abstract:** Sodium-glucose cotransporter-2 (SGLT2) inhibitors are emerging as an important therapy not only for type 2 diabetes (T2DM), but also for heart disease and kidney disease. As these medicines gain acceptance, the number of hospitalized patients receiving them is likely to rise. During clinical trials, SGLT2 inhibitors were noted to have a potential risk for diabetic ketoacidosis (DKA), particularly DKA with relatively normal blood glucose levels, 'euglycemic DKA'. Similar to DKA that is not associated with SGLT2 inhibitors, most of these events seem to be related to acute illnesses or other changes in a patient's medications or self-management circumstances. This creates a need among hospital providers to create strategies to prevent DKA in their hospitalized patient and guidance on monitoring and treating euglycemic DKA. Our combined experience concerning this phenomenon has given a great deal of insight into this problem and the knowledge needed to improve patient care, by augmenting patient education, inpatient surveillance, and early treatment for euglycemic DKA.

**Keywords:** SGLT2 inhibitor; euglycemic diabetic ketoacidosis; hospitalized; peri-operative; type 2 diabetes mellitus

## 1. The Problem

Sodium-glucose cotransporter 2 (SGLT2) inhibitors were introduced to the market in the United States in 2013 as a novel treatment for type 2 diabetes mellitus (T2DM). They function by promoting glucosuria through the inhibition of SGLT2 in the early segments of the proximal tubule, which are responsible for glucose resorption [1]. Since their initial approval, these medications have also been found to reduce hospitalizations for heart failure and cardiac death, as well as reducing the progression of chronic kidney disease [2–4]. These medications are marketed as canagliflozin, dapagliflozin, and empagliflozin, and some are available in dual or triple combinations with metformin and dipeptidyl peptidase enzyme inhibitors (DDP4), respectively. The same agents are approved for the treatment of patients with type 2 diabetes in the European Union. Ipragliflozin is also approved in Asia for the co-management of type 1 diabetes in combination with insulin.

Shortly after coming to the market in the United States, the Food and Drug Administration issued a warning about an increased risk of diabetic ketoacidosis (DKA) in patients taking these medications, based on a handful of incidents identified through the adverse event reporting system [5]. Since then, there have been many case reports and reviews looking into the true incidences and risks of both DKA and euglycemic DKA in the population [6,7]. A large literature review in 2016 showed overall low rates of euglycemic DKA, with the incidence in clinical trials ranging from 0.16 to 0.76 events per 1000 patient years [7]. This and subsequent studies have identified patient characteristics that have been linked with developing euglycemic DKA, including insulin dependence and autoimmune diabetes, which is misdiagnosed as type 2 diabetes [7]. These events were also most often noted in patients who were experiencing another primary acute illness, similar to the triggers recognized for traditional DKA [8,9]. In hospitalized patients, euglycemic DKA

was associated with longer hospital stays and more critical illnesses, though there does not seem to be any differences in the overall mortality rates, based upon small studies and one systematic review [6,10,11].

Given the rising ubiquity of these medications, it is important for hospitalists to be comfortable when caring for hospitalized patients who are on SGLT2 inhibitors. The current guidelines from the Endocrine Society and the American Diabetes Association advise providers to stop giving SGLT2 inhibitors to acutely hospitalized patients and to consider the individuals risks of DKA [12,13]. The Society of Hospital Medicine also recommends a discontinuation, and offers examples of high-risk presentations, including sepsis and hypovolemia [14]. Most surgical societies recommend holding SGLT2 inhibitors 3–5 days before planned procedures, and the Medicines and Healthcare products Regulatory Agency of the United Kingdom recommends testing blood ketones in asymptomatic perioperative patients [15]. To address the growing need of clinicians to be able to manage the risk of euglycemic DKA in hospitalized patients, it is imperative to address the prevention, identification, and treatment of this clinical entity.

## 2. Prevention

There are several ways that a patient's risk of developing euglycemic DKA can be mitigated before and after hospitalization. Physicians prescribing an SGLT2 inhibitor should consider a patient's risk factors for developing euglycemic DKA, including prior episodes of DKA, a ketogenic diet, possible type 1 diabetes, and whether they currently require insulin (Figure 1). It is also important to counsel patients about taking this medication with certain illnesses, particularly those that cause dehydration and decreased oral intake, such as gastroenteritis. Patients should also be counseled about certain activities, such as intense exercise (i.e., a marathon) or excessive alcohol consumption, which increase the risk of euglycemic DKA. Patients should also be aware of the symptoms of DKA and know how to present to the emergency department if they have these symptoms [4].

Prevention can also occur once a patient is hospitalized. The current guidelines stress stopping SGLT2 inhibitors in patients who are hospitalized for acute illnesses or surgery [12–15]. This practice may begin to shift with time, given new data supporting the benefits of these medications in patients with heart failure, with some cardiologists continuing a patient's SGLT2 inhibitor when hospitalized for acute exacerbations [16,17]. These medications are also being initiated or re-started in patients who are hospitalized with heart failure, as part of the guideline's directed medical therapy prior to discharge from the hospital [2,18–20]. Clinicians may need to develop a more nuanced approach to managing these medications, but certain conditions seem to be consistently related to poor outcomes, including sepsis, hypovolemia, diarrheal illnesses, and intolerance of enteral nutrition [4,14].

One trend that has been seen in the literature is for patients who have had significant reductions in their insulin when starting an SGLT2 inhibitor to subsequently develop euglycemic DKA [21]. This is an important consideration for the hospitalized patient, as there are often significant changes made to their insulin regimens while they are an inpatient. Most problematically is the reduction or holding of insulin doses for patients with normal blood glucoses due to reasonable concerns about hypoglycemia. One of the unique challenges of euglycemic DKA is that it can present with a relatively normal glucose in a patient who is still profoundly insulinopenic. While the glucosuria induced by the SGLT2 inhibitor can provide a reassuring blood glucose, the other processes found during classic DKA can still be ongoing, including ketosis, lipolysis, and osmotic diuresis, leading to anion gap metabolic acidosis and profound dehydration.

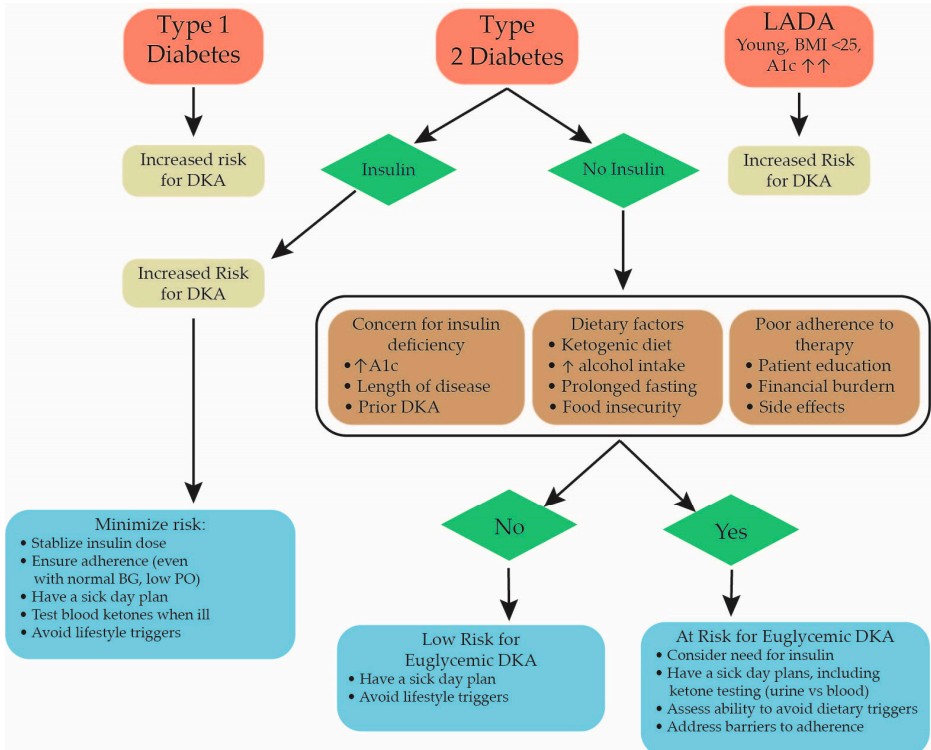

**Figure 1.** Flow chart showing factors to consider when determining a patient's risk for developing euglycemic DKA when starting an SGLT2 inhibitor. Strategies to help minimize a patient's risk based on their unique characteristics are also discussed [7,8]. DKA = diabetic ketoacidosis, A1c = hemoglobin A1c, BG = blood glucose.

Avoiding this will require the intentional management of a patient's insulin regimen to ensure that adequate basal insulin is provided while avoiding hypoglycemia. For patients who were on an SGLT2 inhibitor and insulin before hospitalization, providers should be careful if they are only using sliding scale insulin, as this could lead to an insulinopenic patient receiving no insulin due to normal blood glucose. Some patients who are deemed a high risk for DKA, or are showing early laboratory evidence of DKA, may need supplemental dextrose to allow the administration of insulin while avoiding hypoglycemia. Critically ill patients who are not tolerating oral intake may benefit from an easily titratable intravenous insulin regimen that is structured and monitored similarly to a DKA protocol, to prevent acidosis and hypoglycemia.

### 3. Identification

The diagnosis of euglycemic DKA should be suspected in patients who are taking an SGLT2 inhibitor and who have an elevated or rising anion gap or a low or dropping serum $HCO_3$, regardless of their blood sugar (hence euglycemic, which indicates a blood glucose < 250 mg/dL). Confirmation of the diagnosis is obtained through a venous or arterial blood gas that confirms acidosis, and an identification of ketones, with a blood ketone level preferred to urine due to it having a higher sensitivity (Table 1) [15,22,23]. Suspicion should be especially high in patients presenting with classic DKA symptoms of nausea, vomiting, abdominal pain ,and deep labored respirations, but as these patients are often presenting with some other precipitating illness, identifying these symptoms in relation to DKA is sometimes difficult and delayed.

**Table 1.** Diagnosis of euglycemic DKA requires the presence of euglycemia, and anion gap metabolic acidosis and ketonemia [23]. * Beta-hydroxybutyrate is a sensitive measurement of ketones in DKA. Positive urine ketones can also be used [22].

| Diagnosis of Euglycemic DKA | |
| --- | --- |
| Blood Glucose | <250 mg/dL (13.9 mmol/L) |
| Acidic pH (ABG) | <7.3 |
| Low Serum HCO$_3$ | <18 mEq/L |
| Ketones (Serum BHOB *) | >3 mmol/L |
| Elevated Anion Gap | >10 |

Patients may not have euglycemic DKA on presentation but may develop it during their hospitalization [8,9]. A high level of vigilance can lead to earlier identification of this entity, leading to a quicker initiation of therapy. Daily metabolic panels showing the serum bicarbonate and an anion gap are important, but certain patients may require more frequent monitoring based on their underlying risk factors and severity of their presentation. Such patients may include insulin-dependent diabetics whose insulin doses have been significantly decreased on admission, patients who have had DKA in the past, those who have not been able to eat, and those who are critically ill. Patients who have undergone surgery, particularly abdominal surgery, which often precludes oral intake longer than other procedures, are particularly at risk, and should be closely monitored while not receiving nutrition [8,9,15].

## 4. Management

Once identified, the management of euglycemic DKA is similar to the management of typical DKA. Institutional DKA protocols can be used, but they may require adjustments of any insulin titration parameters that are based on blood glucose. As noted before, these patients have normal blood glucose, which complicates insulin infusion, and they may need higher than typical infusions of dextrose fluids to prevent hypoglycemia while waiting for the acidosis to resolve.

After the initial resolution of their acidosis, if the patient tolerates a diet and is otherwise stable, they should be transitioned to an adequate subcutaneous insulin regimen. This may mean restarting their home regimen, but for insulin-naïve patients, this may be more complicated. In either case, it may be reasonable to assess the continued resolution of the acidosis with a metabolic panel around four hours after transitioning to a subcutaneous insulin plan. In insulin-naïve patients, expert consultation with endocrinology may be indicated to determine whether insulin should be continued on discharge. In some of these patients, it may be that they were already severely insulin deficient and ketosis-prone, and will require ongoing insulin therapy, regardless of whether they continue to take an SGLT2 inhibitor.

The current practice has generally favored the discontinuation of an SGLT2 inhibitor after a patient has experienced euglycemic DKA, but given the growing evidence that these medicines can reduce mortality in both heart and kidney disease, a more nuanced approach may be necessary in the future [2–4]. There are several factors to consider when evaluating the safety of restarting an SGLT2 inhibitor after a patient has had euglycemic DKA. The factors considered when initially prescribing this medication (Table 1) are again applicable. Patients who appear to have an increased baseline risk that is not easily addressed may not be good candidates for re-initiation. If it is determined that a patient now requires insulin therapy, and has been stabilized on a dose, they may actually be more tolerant of having their SGLT2 inhibitor restarted, particularly if they were taking it for heart failure or kidney disease [7]. The circumstances of their admission are also important to help decide if restarting an SGLT2 inhibitor is reasonable. When there are clear insults such as major surgery, infection, or other triggering illnesses that are unlikely to be routine issues for a

patient, they will likely do well restarting the medicine with a clear sick day guidance and vigilance from physicians who care for them during future hospitalizations [4,7].

## 5. Conclusions

Sodium-glucose cotransporter-2 inhibitors are becoming an important tool for treating T2DM, heart disease, and kidney disease, and they have improved the outcomes in many of these patients [2,3]. Like any medication, these medicines come with potential risks, with euglycemic diabetic ketoacidosis being a rare but important complication in this class [6,7,10]. Despite this, there is emerging evidence to support the safety of continuing these medications in carefully selected inpatients, including hemodynamically stable patients admitted with COVID-19 or those with an acute heart failure exacerbation [18,20,24,25].

The risk of developing euglycemic DKA can be reduced by carefully selecting patients for therapy and providing clear counseling about the signs and triggers of euglycemic DKA. Providers should take care to actively monitor for this complication in all hospitalized patients who are prescribed this medication. If a patient is maintained on their SGLT2 inhibitor during hospitalization, it should be discontinued promptly if they become hemodynamically unstable, unable to tolerate enteral nutrition, or if they show laboratory evidence of developing metabolic acidosis (Table 1). Patients who were previously on insulin should not have the therapy completely discontinued and should not be kept solely on a sliding scale insulin dose.

The management of euglycemic DKA is like that of typical DKA, with the knowledge that their glucose levels may not be particularly elevated. Afterwards, restarting an SGLT2 inhibitor should involve an assessment of a patient's future risk for euglycemic DKA and shared decision-making and education about the future risks and ways to mitigate it.

| Practice Points |
| --- |
| Maintain a high suspicion for euglycemic DKA in patients taking SGLT2 inhibitors with symptoms or laboratory abnormalities seen in DKA, regardless of blood glucose. |
| Be intentional when ordering insulin for a patient who has taken SGLT2 inhibitors; dose reductions increase the risk of DKA. |
| Carefully consider continuing insulin at discharge in an insulin-naïve patient presenting with euglycemic DKA. |
| Discuss risk factors and signs of DKA with all patients before starting an SGLT2 inhibitor, and provide a clear sick day plan for their insulin and SGLT2 inhibitors treatment regime. |

**Author Contributions:** Conceptualization, J.S.; Writing- original draft preparation, J.S.; writing-review and editing, J.S., A.R.P.-L.; visualization, J.S.; supervision, A.R.P.-L. All authors have read and agreed to the published version of the manuscript.

**Funding:** This research received no external funding.

**Institutional Review Board Statement:** Not applicable.

**Informed Consent Statement:** Not applicable.

**Data Availability Statement:** Not applicable.

**Acknowledgments:** No funding support was available for this work. A.R.P.-L. is supported by an NIDDK grant R01DK130992-01.

**Conflicts of Interest:** The authors declare no conflict of interest.

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
