# Peer review of "Managing Hospitalized Patients Taking SGLT2 Inhibitors: Reducing the Risk of Euglycemic Diabetic Ketoacidosis"

_diabetology, doi:10.3390/diabetology4010010_

Round 1

Reviewer 1 Report

The major remit of this mini-review is to tell that number of patients on SGLT2 inhibitors will increase, and also that there are reasons to keep the patients on SGLT2 inhibitors while hospitalized for e.g. heart failure, and start treatment with SGLT2 inhibitors during the acute phase of the disease will be more common. I agree on this, and I also agree also to avoid reduction in insulin dose, if possible, and to frequently monitor anion gap and bicarbonate levels when the patients are acutely ill. A study that could be referred to is Kosiborod et al. 2021  (Dapagliflozin in patients with cardiometabolic risk factors hospitalised with COVID-19 (DARE-19): a randomised, double-blind, placebo-controlled, phase 3 trial. Lancet Diabetes Endocrinol. 2021 Sep;9(9):586-594), where they investigated SGLT2 inhibitor treatment in acutely ill patients hospitalized for COVID-19.

In section 1, the first paragraph, lines 24-35, content and references need to be updated. Line 27: SGLT2 are co-transporters, so take out “co-transporters”, line 30: reference 2-4 are not appropriate. Please refer to the original studies. It is not correct to say that mortality in heart failure is reduced; hospitalization for heart failure and cardiovascular death was reduced.   Line 33-35: includes typos and not correct information. Line 33-34 should read “…in the European Union.” Please check if the mentioned SGLT2 inhibitors on line 34 are approved for type 1 diabetes. I don’t think dapagliflozin is any longer approved for treatment of type 1 diabetes. I think the application was withdrawn.

In section 3, please add, preferably a table the definition of DKA, including plasma levels of ketones. This is to illustrate the information line 120-123. Line 121: what is meant by anion gap metabolic acidosis? The anion gap can indicate metabolic acidosis, or the pH show that there is acidosis. Please correct.

In section 5, I miss a summary and conclusion when the authors recommend circumstances when to keep SGLT2 inhibitor treatment while hospitalized for acute illness and circumstances when they would recommend to hold treatment. Also clarify what is meant by “…if clinical status changes.” line 186-187.

Author Response

Thank you for your feedback, below are the edits I have made based on your recommendations:

  • I changed the wording regarding an increase in patients taking SGLT2i
  • I provided more refs and details on patients who may benefit from continuing the med as inpatients (also in section 5)
  • Removed second instance of co-transporter from line 27
  • reviewed references 2-4 and updated wording accordingly (hf hospitalizations and cardiac death instead of mortality)
  • Dapa is no longer approved for T1D, so removed this line
  • Table added to section 3 utilizing ADA guidelines for lab diagnosis of mild DKA then incorporating euglycemia
  • More detail provided about AGMA in the text as well as in the table
  • Section 5 included more specific examples of patients who could continue, I think the clarification I put in about change in clinical status should suffice for the population that should not continue?

Reviewer 2 Report

Very important topic, given the use of SGLT2 is increasing given its approval for Heart failure and Diabetes. 

1. I see the term Diabetes everywhere, but not T2DM. Are you suggesting use of SGLT-2s for T1DM, LADA and T2DM, as currently its only approved for DM2, heart failure and CKD ! 

2. In your "The problem section, last paragraph", that sentence seems redundant and similar to the 1st line of the paragraph. please consider changes so they don't too similar and redundant. 

3. In the "Identfication" paragraph, Do you have data on number of patients in the trials who had DKA while inpatient. would be good to know some stats about the gravity of the situation. 

4. Have you thought of meta-analyses or systematic review of these trials ? so far this paper seems like a mini-review, and it does a good job ! 

Author Response

Thank you for your feedback and encouragement, below are the changes I made based on your review:

  • References to just diabetes have been replaced with T2DM
  • First and last sentence of the final paragraph of section 1 have been edited to reduce the redundancy
  • Apart from the stats of frequency in the first section, the data specifically looking at developing eDKA while hospitalized is scant. One of my refs looks at all patients admitted with DKA and compares characteristics of those on SGLT2 vs not, and they do find that the SGLT2 group is more likely to develop DKA while hospitalized instead of presenting with the diagnosis, I'm just not sure how to present this info in a meaningful way in this text
  • Moving onto a full meta vs lit review of developing eDKA as an inpatient would be pretty interesting, I'm not sure enough sources make the distinction to make a full review feasible, but I'll think about it